# The Effect of Microbiome-Modulating Agents (MMAs) on Type 1 Diabetes: A Systematic Review and Meta-Analysis of Randomized Controlled Trials

**DOI:** 10.3390/nu16111675

**Published:** 2024-05-29

**Authors:** Ying Zhang, Aiying Huang, Jun Li, William Munthali, Saiying Cao, Ulfah Mahardika Pramono Putri, Lina Yang

**Affiliations:** 1Xiangya School of Public Health, Central South University, Changsha 410128, China; 236911029@csu.edu.cn (Y.Z.);; 2Department of Chemistry, University of Washington, Seattle, WA 98195, USA; 3School of Psychology, South China Normal University, Guangzhou 510631, China

**Keywords:** type 1 diabetes, randomized controlled trial, probiotic, prebiotic, postbiotic, synbiotic, meta-analysis

## Abstract

Gut microbiome-modulating agents (MMAs), including probiotics, prebiotics, postbiotics, and synbiotics, are shown to ameliorate type 1 diabetes (T1D) by restoring the microbiome from dysbiosis. The objective of this systematic review and meta-analysis was to assess the impact of MMAs on hemoglobin A1c (HbA1c) and biomarkers associated with (T1D). A comprehensive search was conducted in PubMed, Web of Science, Embase, Cochrane Library, National Knowledge Infrastructure, WeiPu, and WanFang Data up to 30 November 2023. Ten randomized controlled trials (*n* = 630) were included, with study quality evaluated using the Cochrane risk-of-bias tool. Random-effect models with standardized mean differences (SMDs) were utilized. MMA supplementation was associated with improvements in HbA1c (SMD = −0.52, 95% CI [−0.83, −0.20]), daily insulin usage (SMD = −0.41, 95% confidence interval (CI) [−0.76, −0.07]), and fasting C-peptide (SMD = 0.99, 95% CI [0.17, 1.81]) but had no effects on FBG, CRP, TNF-α, IL-10, LDL, HDL, and the Shannon index. Subgroup analysis of HbA1c indicated that a long-term intervention (>3 months) might exert a more substantial effect. These findings suggest an association between MMAs and glycemic control in T1D. Further large-scale clinical trials are necessary to confirm these findings with investigations on inflammation and gut microbiota composition while adjusting confounding factors such as diet, physical activity, and the dose and form of MMA intervention.

## 1. Introduction

Type 1 diabetes (T1D) refers to an autoimmune disease leading to the self-destruction of insulin-producing pancreatic ß cells and insulin deficiency, which leads to impaired glucose metabolism [1]. T1D places heavy burdens on public health due to the rapid increase in prevalence rate and its complex condition for glucose management, especially in resource-limited countries [2]. Insulin therapy is the most accepted treatment for T1D, which requires subcutaneous insulin injection several times per day [3]. It causes several challenges, including high expenses, weight gain, risk of hypoglycemia, and low adherence [4]. Therefore, novel and economic therapy with high adherence and accessibility is needed to slow down the progression of T1D [5].

The gut microbiome has been shown to impact the occurrence and pathogenesis of T1D in recent years [6]. Case–control studies indicate that compared with healthy control subjects, T1D is associated with a significantly lower microbiota diversity, a higher relative abundance of Bacteroides, Ruminococcus, Blautia, and Streptococcus genera, and a lower relative abundance of Bifidobacterium, Roseburia, and Faecalibacterium [7]. An imbalanced Bacteroidetes-to-Firmicutes ratio leads to dysbiosis, which changes intestinal mucosa and alters gut permeability, resulting in a leaky gut [8,9]. In T1D subjects, the disharmonized intestinal microenvironment causes an increased level of proinflammatory cytokines and lipopolysaccharides (LPSs), and they enter into the bloodstream with greater accessibility since the tight junctions between colonocytes are damaged, resulting in an increased level of inflammatory substances in the bloodstream [10]. As a result, the inflammation status causes islet autoimmunity, leading to decreased fasting C-peptide (FCP) and elevated glycemic levels. FCP reflects endogenous insulin production and provides insights into residual beta-cell activity, which is commonly used to assess the effectiveness of interventions aimed at preserving or enhancing insulin secretion [11]. Glycated hemoglobin (HbA1c), a widely used biomarker for assessing long-term glucose control in individuals with diabetes, reflects the average blood glucose levels over the past 2–3 months, providing information about the effectiveness of diabetes management strategies [12].

Tackling dysbiosis is suggested to be a novel strategy for treating T1D, and using MMAs is considered to be a feasible way of restoring the gut microbiota [13]. MMAs are substances that regulate the gut microbiota, including probiotics, prebiotics, synbiotics, and postbiotics. Supplementation with probiotics in T1D adults has shown improved glycemic control and increased synthesis of Glucagon-like peptide-1 [14]. Prebiotics play a role in an increase in the number of lactic acid-producing bacteria and have immuno-modulatory properties [15]. Postbiotics from various microbiomes inhibit the growth of pathogenic bacteria [16]. Synbiotics exert a synergistic effect on the restoration of the gut microbiota [17]. Therefore, MMAs might play a role in maintaining gut microbiota homeostasis, stabilizing blood glucose levels, and reducing the level of proinflammatory cytokines while increasing anti-inflammatory cytokines, resulting in a slower T1D progression [18].

While there are abundant reviews exploring the relationship between MMAs and glycemic control, the majority of the literature predominantly focuses on type 2 diabetes (T2D). Conversely, the literature specifically addressing T1D is notably limited, with only two existing reviews identified. One review, encompassing five randomized controlled trials (RCTs) up to 8 October 2022, examined the impact of probiotics and synbiotics on glycemic control, focusing on outcomes such as fasting blood glucose (FBG), HbA1c, fasting C-peptide (FCP), and daily insulin usage (DIU) [19]. However, it did not delve into the outcomes related to T1D pathogenesis, such as inflammatory cytokines and gut microbiota composition. Another recent review aimed to explore the effects of probiotic and synbiotic interventions on both T1D and T2D [20]. Despite its inclusion of a large overall sample size, individuals with T1D represented only 2.8% (*n* = 84), and the review excluded patients with diabetes under 18, a demographic where T1D is prevalent. Moreover, this review was unable to differentiate the outcomes between T1D and T2D, which is crucial due to their distinct pathophysiologies, treatment modalities, and potential responses to interventions. Consequently, there remains a gap in the literature regarding quantitative review studies on T1D. To address this gap, the current meta-analysis updates the evidence up to 30 November 2023, incorporating ten studies covering children, adolescents, and adults and employing a more comprehensive set of outcome measures.

## 2. Materials and Methods

### 2.1. Data Sources and Literature Search

This meta-analysis was recorded in the International Prospective Register for Systematic Reviews (PROSPERO) under registration number CRD42023395896. The review adhered to the guidelines outlined in the Preferred Reporting Items for Systematic Reviews and Meta-Analyses (PRISMA) statement [21], which can be found in Appendix A, and a comprehensive literature search was undertaken by two independent researchers across seven databases, namely PubMed, Web of Science, Embase, Cochrane Library, National Knowledge Infrastructure (CNKI), WeiPu (VIP), and WanFang Data (WanFang), until 30 November 2023. Published reviews and their references were also manually searched to identify any additional studies meeting the inclusion criteria. A combination of MeSH terms and free text were utilized, encompassing terms such as ‘type 1 diabetes’, ‘probiotics’, ‘synbiotics’, and ‘randomized controlled trials’. Boolean operators were employed for sensitivity (‘OR’) and precision (‘AND’), customized to the syntax of each individual database. As an example, the search methodology applied in PubMed was structured as (‘Diabetes Mellitus, Type 1’[Majr]) AND (‘Probiotics’[Majr] OR ‘Prebiotics’[Majr] OR ‘inulin’[Majr] OR ‘bifidobacterium’[Majr] OR ‘lactoccocus’[Majr] OR ‘butyrate’[Majr]) with a clinical trial filter. The details of the search methodologies employed are provided in Appendix A.

### 2.2. Inclusion and Exclusion Criteria

A study was included if the following criteria were met: (1) RCT; (2) the literature was published before 30 November 2023; (3) the subjects must be diagnosed specifically with T1D; notably, no specific criteria were set for participants’ age or disease duration, aiming to encompass a broad spectrum of eligible studies; (4) interventions were limited to probiotics, synbiotics, prebiotics, and postbiotics with no requirement on duration; and (5) the primary outcome was HbA1c, and the secondary outcomes were FBG, FCP, DIU, C-reactive protein (CRP), interleukin-10 (IL-10), tumor necrosis factor-α (TNF-α), high-density lipoprotein (HDL), low-density lipoprotein (LDL), and Shannon index.

The exclusion criteria were as follows: (1) the subjects had other types of disease; (2) the probiotics were taken within three months before the trial; and (3) duplicate studies.

### 2.3. Selection and Data Extraction Process

Rayyan is a screening tool used for systematic reviews and meta-analyses, facilitating the efficient selection and management of relevant studies [22], and was employed in this review. During the initial round of title and abstract screening, both reviewers independently assessed all 831 records. Subsequently, in the second phase of full-text screening, a panel of two reviewers collectively evaluated the articles. Any disparities or disagreements that emerged during this process were addressed through collaborative discussion between the two reviewers, persisting until unanimous agreements were reached.

Two authors conducted data extraction independently, encompassing key aspects including (1) first author, publication year, and study country; (2) study design and intervention duration; (3) comprehensive details regarding the intervention and placebo, including specific probiotic strain, dosage, and daily intake time; (4) baseline characteristics like age, disease duration, and body mass index (BMI); and (5) metabolic outcomes, which were measured both before and after interventions. The extracted data underwent a verification process by both authors. In instances where data were not explicitly presented in the publications, the data analyst sought information in Appendix A. If the necessary details remained elusive, the corresponding authors were contacted via email to solicit the missing data. A systematic follow-up protocol was implemented: After the initial contact, a one-week interval was allowed for a response. If no reply was received, a second contact attempt was made. In the event of continued non-response after the second attempt, the study was excluded from the analysis.

### 2.4. Quality Assessment

The quality assessment of each RCT was independently conducted by two reviewers utilizing the Cochrane risk-of-bias tool (ROB2) [23]. Additionally, the ROBVIS tool [24] was employed for visualization purposes. Adhering to the ROB2 guidelines, the following biases were systematically assessed: (1) bias arising from the randomization process; (2) bias due to deviations from intended interventions; (3) bias due to missing outcome data; (4) bias in the measurement of the outcome; and (5) bias in the selection of the reported result. The tool automatically synthesized the overall risk of bias, represented as low risk in green, some concerns in yellow, and high risk in red. Any disparities in the assessment were meticulously resolved through collaborative discussions between the two reviewers, persisting until consensus was achieved. The overall certainty of evidence across the studies was graded according to the GRADE (Grading of Recommendations Assessment, Development, and Evaluation) working group guidelines. The quality of evidence was classified into four categories, namely high, moderate, low, and very low, according to the corresponding evaluation criteria.

### 2.5. Data Synthesis and Statistical Analyses

For the synthesis and quantitative analysis of data, Review Manager (Revman) 5.3 software was employed in this study. Continuous data were presented as the mean difference with standard deviation (m ± SD). In cases where data were initially expressed as median with interquartile range (IQR) or range, the skewness was assessed using the website (www.math.hkbu.edu.hk, accessed on 1 April 2024) [25,26]. If the data were not significantly skewed, transformation into mean with SD was undertaken. The standardized mean difference (SMD) with 95% confidence intervals (CIs) was calculated using random-effect models. In random-effect models, the treatment effect estimates observed in studies may vary due to genuine disparities in treatment effects across each study, along with sampling variability. This diversity in treatment effects could be attributed to discrepancies in study populations (e.g., patient age), interventions administered (e.g., drug dosage), duration of follow-up, and other variables. Thus, a random-effect model was utilized by facilitating the extension of findings beyond the included studies by assuming that these studies represented random samples from a broader population. Statistical significance was established at *p* < 0.05. Heterogeneity was evaluated through *I*^2^, and *I*^2^ values of 25%, 50%, and 75% were suggested to be indicators of low, moderate, and high heterogeneity, respectively [27]. Sensitivity analyses were performed for results displaying high heterogeneity to assess whether the combined outcomes and heterogeneity altered, aiming to evaluate the robustness of the findings. Subgroup analyses were further performed based on the different MMAs used, age, disease duration, and intervention duration differences. In cases where more than 10 studies were included, potential publication bias was investigated utilizing funnel plots [28].

## 3. Results

### 3.1. Literature Search Results

This review initially identified 831 records, of which 132 were excluded due to duplication. During the process of screening titles and abstracts, 680 studies were eliminated, primarily on the basis of irrelevant diseases, including type 2 diabetes (T2D), gestational diabetes mellitus (GDM), and latent autoimmune diabetes in adults (LADA); non-human studies including in vivo and in vitro studies; non-interventional studies, such as cross-sectional, cohort, and case–control studies; and reviews and protocols. In total, 19 full articles were reviewed for eligibility, and eventually, 10 clinical trials were included with 630 patients’ records in this meta-analysis (Figure 1). The exclusion reasons for the other nine articles are indicated in Supplementary S2.

### 3.2. Basic Characteristics of the Included Studies

Table 1 Summarizes essential data from the 10 included RCTs. One trial included children with three age groups [29]. Thus, each age group was considered as an individual report, and 12 subgroups were obtained eventually. A total of 630 participants (315 in the intervention group and 315 in the control group) underwent re-analysis. All clinical trials included both genders, maintaining a balanced male-to-female ratio (1.02).

Regarding interventions, one study used a prebiotic (inulin) [13], two studies used a postbiotic (sodium butyrate) [30,31], one study used a single-strain probiotic (*Lactobacillus rhamnosus* GG) [32], and six used multistrain probiotic or synbiotic supplements [29,33,34,35,36,37]. To maximize inclusivity, eligibility criteria did not impose restrictions based on age or the duration of T1D. Planned subgroup analyses considered age differences and disease duration. Mean participant age ranged from 5.5 to 56 years, and T1D duration ranged from 1.5 to 30 years. Six studies implemented MMA supplementation for 3 months [30,31,32,33,35], with one study using less than 3 months [34] and three studies exceeding 3 months [29,34,37]. Beyond post-intervention assessments, three studies also measured outcomes 3 months and 6 months after intervention completion [15,34,37]. None of the trials reported any significant adverse events in the MMA intervention group.

**Table 1 nutrients-16-01675-t001:** Characteristics of 10 RCTs that investigated the effect of MMAs on T1D.

Author/Year	Country	Sample Size	Age †	T1D Duration (Year)	Intervention; Control	Dose	Follow-Up	Main Measured Biomarkers
Bianchini et al., 2020 [30]	Italy	Probiotic: 34	13.4 (4.67)	NA	Probiotic drop: *Lactobacillus rhamnosus* GG	5 × 10^9^ LGG/drops, BID	3 mo	HbA1c
		Control: 30	13.1 (4.7)		Placebo drop: with a similar formulation but not containing probiotic	5 drops, BID		
Groele et al., 2021 [37]	Poland	Probiotic: 48	12.3 (2.13)	<2 months	Multistrain probiotic capsule: *L. rhamnosus* GG ATCC 53103 and *B. lactis* Bb12 DSM 15954	10^9^ CFU/capsule, QD	6 mo; 6 mo AI	HbA1c, AUC_C-peptide_, FCP, DIU, IL-10, TNF-α
		Control: 48	13.17 (2.59)		Placebo capsule: maltodextrin	1 capsule, QD		
Groot et al., 2020 [29]	The Netherlands	Postbiotic: 50	32.5 (22–61) ‡	8 (4–16)	Postbiotic capsule: sodium butyrate	2 g/capsule, TID	3 mo	HbA1c, DIU, FCP, CRP, LDL, Fecal SCFA
		Control: 50	32.5 (22–61) ‡		Placebo capsule	2 g/capsule, TID		
Ho et al., 2019 [15]	Canada	Prebiotic: 17	12.52 (2.76)	7.31 (3.93)	Prebiotic capsule: oligofructose enriched inulin (chicory root-derived)	8 g/capsule, QD	3 mo; 3 mo AI	HbA1c, FCP, gut microbiota
		Control: 21	11.94 (2.61)	4.70 (3.07)	Placebo capsule: maltodextrin	3.3 g/capsule, QD		
Javid et al., 2020 [36]	Iran	Synbiotic: 22	10.36 (2.53)	4.45 (1.96)	Synbiotic powder: *Lactobacillus sporogenes* GBI-30, maltodextrin, and fructooligosaccharide	2 g powder (10^9^ CFU), QD	2 mo	HbA1c, FBG, FCP, DIU, HDL, LDL, CRP
		Control: 22	10.04 (2.08)	4.04 (1.36)	Placebo powder: starch	2 g powder, QD		
Kumar et al., 2021 [33]	India	Probiotic: 47	7.92 (3.92)	<2 months	Multistrain probiotic capsule: *L. paracasei* DSM 24733, *L. plantarum* DSM 24730, *L. acidophilus* DSM 24735, and *L. delbrueckii* subsp. bulgaricus DSM 24734, *B. longum* DSM 24736, *B. infantis* DSM 24737, *B. breve* DSM 24732, and *Streptococcus thermophilus* DSM 24731	1.125 × 10^11^ bacteria/capsule, QD	3 mo	HbA1c, FBG, FCP, DIU
		Control: 49	9.1 (4.95)	<2 months	Placebo capsule: microcrystalline cellulose	1 capsule, QD		
Lin et al., 2021 [29]	China	Probiotic: 35	1–15 (3 age groups)	>1 year	Multistrain probiotic capsule: *Bifidobacterium longum*, *Lactobacterium bulagricumi*, and *Streptococcus thermophilus*	4.2 × 10^7^ CFU/capsule, 2–4 TID (based on age)	6 mo	HbA1c, FCP, DIU CD4+/CD8+
		Control: 35			Placebo: insulin therapy			
Tougaard et al., 2022 [30]	Finland	Postbiotic: 28	56 (11)	29 (17)	Postbiotic granules: sodium butyrate	1.8 g, TID	3 mo	HbA1c, fecal SCFA
		Control: 25	52 (15)	32 (14)	Placebo capsule: microcrystalline cellulose	QD		
Wang et al., 2022 [34]	Taiwan, China	Probiotic: 27	14.1 (5.1)	6.2 (4.5)	Multistrain probiotic capsule: 1:1 mixture ratio of *Lactobacillus salivarius* subsp. salicinius AP-32, *L. johnsonii* MH-68, and *Bifidobacterium animalis* subsp. lactis CP-9	5 × 10^9^ CFU/capsule, QD	6 mo; 3 mo AI	HbA1c, FBG, TNF-α, gut microbiota
		Control: 29	14.3 (4.6)	6.4 (4.1)	Placebo: insulin therapy			
Zhang et al., 2023 [35]	China	Probiotic: 27	38 (14)	10 (4, 16)	Multistrain probiotic capsule: *Bifidobacterium longum*, *Lactobacterium bulagricumi*, and *Streptococcus thermophilus*	4.2 × 10^7^ CFU/capsule, TID	3 mo	HbA1c, FBG, FCP, DIU, LDL, HDL CGM
		Control: 23	39 (8)	10 (7, 16)	Placebo capsule: same substances but without the bacteria	1 capsule, TID		

† Note: normally distributed quantitative variables are presented as the mean ± SD. ‡ Non-normally distributed quantitative variables are presented as the median (interquartile range, IQR). Functional abbreviations: CFU, colony forming unit; QD, once a day; TID, three times a day; CGM, continuous glucose monitoring; mo, months; AI, after intervention, meaning the study included a follow-up for a period of time when intervention completed. Study outcome abbreviations: HbA1c, glycated hemoglobin; AUC_C-peptide_, area under the curve of the C-peptide level during 2 h responses to a mixed meal; FCP, fasting C-peptide; DIU, daily insulin usage; TNF-α, tumor necrosis factor-α; FBG, fasting blood glucose; SCFA, short-chain fatty acid; CRP, C-reactive protein; HDL, high-density lipoprotein; LDL, low-density lipoprotein; NA, Data not applicable; BID, twice a day.

### 3.3. Risk-of-Bias Assessment

The results of Cochrane’s risk-of-bias assessment in Figure 2 show that the overall risk of bias was of some concern. All studies described the generation of random sequences in detail, six studies reported proper allocation concealment [15,30,31,32,33,37], and six studies demonstrated the integrity of data [15,30,31,32,34,37]. One study had incomplete results due to missing data [32]. In seven studies, it was unclear whether there was reporting bias [29,30,32,33,34,36,37].

### 3.4. Meta-Analysis Results

#### 3.4.1. Effects of MMA Intervention on HbA1c

The effect of MMAs on HbA1c was reported by 10 studies (*n* = 600) [29,30,32,33,34,36,37], as depicted in Figure 3. The overall effect (SMD = −0.52, 95% CI [−0.83, −0.20], *p* < 0.01) indicated a significant improvement in HbA1c with MMA intervention but with moderate heterogeneity (*I*^2^ = 70%, *p* < 0.01, Tau^2^ = 0.18). The sensitivity analysis revealed that the omission of any single study did not significantly alter the result. Subgroup analyses were conducted based on four parameters, namely the duration of intervention, the MMAs used, age, and the disease duration of the T1D patients, which are discussed in Section 3.4.5.

#### 3.4.2. The Effect of MMA Intervention on Daily Insulin Usage

The effect of MMAs on DIU was reported by three studies with five subgroups (*n* = 250) [29,33,37] and a significant decrease in usage was found (SMD = −0.41, 95% CI [−0.76, −0.07], *p* = 0.02); the heterogeneity was moderate (*I*^2^ = 35%, *p* = 0.19, Tau^2^ = 0.05) (Figure 4).

#### 3.4.3. The Effect of MMA Intervention on Fasting C-Peptide

The effect of MMAs on FCP was reported by five studies with seven subgroups (*n* = 336) [15,29,33,35,37], and the reports indicated a significant improvement (SMD = 0.99, 95% CI [0.17, 1.81], *p* = 0.02) (Figure 5). Sensitivity analysis was performed due to high heterogeneity (*I*^2^ = 90%, *p =* 0.01, Tau^2^ = 1.00). The omission of any single study or subgroup did not significantly alter the result; thus, subgroup analyses were performed.

#### 3.4.4. The Effect of MMA Intervention on Other Results

FBG was reported in three studies (*n* = 154) [34,35,36], and the result was not significant (SMD = −0.29, 95% CI [−0.62, 0.03], *p* = 0.08). CRP was reported in three studies (*n* = 192) [30,35,36], and the result was not significant (SMD = −0.25, 95% CI [−0.84, 0.33], *p* = 0.40). TNF-α was reported in three studies (*n* = 184) [13,32,35], and the result was not significant (SMD = −0.03, 95% CI [−0.58, 0.52], *p* = 0.91). IL−10 was reported in two studies (*n* = 128) [15,37], and the result was not significant (SMD = 0.31, 95% CI [−0.04, 0.66], *p* = 0.08). There were two studies (*n* = 92) that investigated HDL (SMD = 0.27, 95% CI [−0.25, 0.80], *p* = 0.31) and LDL (SMD = −0.23, 95% CI [−0.98, 0.51], *p* = 0.54), and both results were insignificant [35,36]. The Shannon index was reported in two studies (*n* = 68) [15,31], and there was no difference (SMD = −0.66; 95% CI [−1.62, 0.30], *p* = 0.18). All the results above are shown in Appendix A. Regarding adverse events, none of the included studies reported any occurrences.

The efficacy of MMAs as evidenced by the post-interventional results of HbA1c was reported by three studies (*n* = 367) [15,34,37]. After aggregating the outcome at the end of the intervention (SMD = −0.74, 95% CI [−1.34, −0.41], *p* = 0.02), and after 3 months (SMD = −0.18, 95% CI [−0.47, 0.11], *p* = 0.22), a diminished effect was observed, as shown in Figure 6.

#### 3.4.5. Subgroup Analysis

##### Influence of Intervention Duration on HbA1c

The subgroup analysis involving the intervention duration demonstrated a considerable impact in the subgroup of more than 3 months (SMD = −1.06, 95% CI [−1.37, −0.75], *p* < 0.01) [29,34,37] and an insignificant effect in the other subgroup (SMD = −0.17, 95% CI [−0.37, 0.04], *p* = 0.11) [15,30,32,33,34,36], as shown in Figure 7. Heterogeneity rates in both subgroups were low (8% for more than 3 months, and 0% for equal or less than 3 months), and the test for subgroup differences indicated a statistically significant effect (Chi^2^ = 22.74, *df* = 1 (*p* < 0.001), *I*^2^ = 95.6%).

##### Influence of Age on HbA1c

Subgroup analysis involving age showed a significant effect (SMD = −0.59, 95% CI [−1.00, −0.19], *p* < 0.01) in the children and adolescent groups [15,29,32,33,36], whereas this was not observed in adults (SMD = −0.27, 95% CI [−0.60, 0.05], *p* = 0.10) [28,33], as shown in Figure 8. Heterogeneity was considerable in the children subgroup (*I*^2^ = 74%), and very low among adults (*I*^2^ = 0%). In addition, there was no significant difference between these two subgroups (Chi^2^ = 1.46, *df* = 1 (*p* = 0.23), *I*^2^ = 31.7%).

##### Influence of Different MMAs on HbA1c

The subgroup analysis involving different MMAs revealed a distinct pattern between subgroups (Chi^2^ = 5.66, *df* = 1 (*p* = 0.02), *I*^2^ = 82.3%), as shown in Figure 5. The intention was to group each MMA, but studies on single-strain probiotics [32], prebiotics [15], and postbiotics [30] only involved one group, not allowing for separate grouping; thus, two subgroups were established. Studies on multistrain probiotics and synbiotics [29,33,34,35,36,37] revealed significant effects (SMD = −0.71, 95% CI [−1.08, −0.34], *p* < 0.01), whereas an insignificant improvement was found in another subgroup (SMD = −0.09, 95% CI [−0.44, 0.27], *p* = 0.63) (Figure 9). Heterogeneity was considerable in the multispecies subgroup (*I*^2^ = 35) and not significant in the monospecies subgroup (*I*^2^ = 35%).

##### Influence of Disease Duration of MMAs on HbA1c

Subgroup analysis by disease duration showed a significant improvement in long-term T1D (SMD = −0.43, 95% CI [−0.75, −0.11], *p* < 0.01) [29,33,34,35,36,37] and an insignificant effect on onset T1D (SMD = −0.75, 95% CI [−1.77, 0.26], *p* = 0.15) [33,37], as depicted in Figure 10. Heterogeneity was considerable in the onset-T1D subgroup (*I*^2^ = 91%) and moderate in the long-term-T1D subgroup (*I*^2^ = 35%). Additionally, an in-between difference was not observed in these two subgroups (Chi^2^ = 0.35, *df* = 1 (*p* = 0.56), *I*^2^ = 0%).

##### Influence of Age, MMAs, and Disease Duration on Fasting C-Peptide

Subgroup analyses involving FCP are presented in Appendix A. One study included participants more than 18 years old and showed no effect (SMD = 0.05, 95% CI [−0.52, 0.61], *p* = 0.88); individuals with an age less than or equal to 18 years showed a significant improvement (SMD = 1.21, 95% CI [0.20, 2.21], *p* = 0.02). The results of multistrain probiotics and synbiotics showed a significant improvement in FCP (SMD = 1.06, 95% CI [0.09, 2.02], *p* = 0.03). The other group contained one study and indicated an improvement (SMD = 0.75, 95% CI [0.09, 1.42], *p* = 0.03). There was no significant improvement among individuals with disease duration less than or equal to 1 year (SMD = −0.15, 95% CI [−0.44, 0.15], *p* = 0.32), whereas significant improvement was observed among individuals with disease duration more than 1 year (SMD = 1.61, 95% CI [0.37, 2.86], *p* = 0.01).

### 3.5. Publication Bias

Publication bias was assessed for HbA1c since it was the only biomarker that exceeded 10 studies and subgroups. The plot shows a concentrated distribution of all studies, suggesting an absence of publication bias (Appendix A).

### 3.6. Grading of Evidence

An evaluation of the quality of evidence using the GRADE approach is presented in Table 2. The quality of evidence was moderate for HbA1c due to inconsistency (*I*^2^ = 70%), low quality for FCP, and very low quality for DIU, owing to limitations on imprecision (*n* = 336 and *n* = 250 for sample size, respectively), as well as limitations on inconsistency for FCP (*I*^2^ = 90%) and risk of bias in DIU and publication bias (two studies had an overall uncertain risk, and one was with high risk of bias).

## 4. Discussion

This systematic review and meta-analysis focused on probiotics, prebiotics, synbiotics, and postbiotics as adjuvant therapy in T1D management. With the inclusion of 10 clinical trials, comprising a sample size of 630 T1D patients, significant improvements were observed in HbA1c, FCP, and DIU, while no effects were found in FBG, CRP, TNF-α, IL-10, HDL, LDL, and the Shannon index. Subgroup analyses based on HbA1c revealed the effects of the intervention period, types of MMAs, age, and the disease duration of the patients. A considerable effect on HbA1c was found in the subgroup receiving multistrain probiotics or synbiotics, a supplementation period for more than 3 months, and in patients under 18 years old with long-term T1D. The grading of the quality of evidence indicated moderate quality of evidence in HbA1c and low/very low quality of evidence in FCP and DIU.

Comprising roughly 1000 species and weighing approximately 1.5 kg, the gut microbiota is integral to human health, and alterations in its composition, known as dysbiosis, have been implicated in the pathogenesis of T1D [38]. The gut microbiome composition was different in healthy versus TID in both human and animal models [39]. Animal studies indicated higher alpha-diversity in the gut microbiota of non-obese diabetic (NOD) mice compared with mice that later progressed to T1D [40], and in mice with reduced T1D progression, a higher Bacteroidetes-to-Firmicutes ratio was observed [41]. Similar to animal models, case–control studies indicated a significantly lower Bacteroidetes-to-Firmicutes ratio in the T1D group [7,8]. The “Teddy study (The Environmental Determinants of Diabetes in the Young)” showed a lower abundance of *Streptococcus thermophilus* and *Lactococcus lactis* in children at the onset of T1D with respect to healthy subjects [42]. In addition, children with T1D were observed with decreased numbers of bacteria that were essential to maintain gut integrity such as lactic acid-producing bacteria, butyrate-producing bacteria, and mucin-degrading bacteria. Aberrant gut microbiota composition might play a pivotal role in the development of T1D mainly by modulating the formation of SCFA [43], compromising the gut barrier by loosening the tight junction between cells, allowing pathogenic substances such as TNF-α to enter the bloodstream, and triggering autoimmune responses underlying T1D [44].

The modulation of the gut microbiota is a strategy aiming at reversing dysbiosis by using different types of MMAs [45]. Single-strain probiotics, multistrain probiotics, synbiotics, prebiotics, and postbiotics were included in this review, with multistrain probiotics appearing to exert a greater efficacy, aligning with the literature [46]. Treating NOD mice with probiotic strains belonging to families Bifidobacteriaceae and Lactobacillaceae and the *Streptococcus thermophilus* genus has been shown to ameliorate T1D [47]. The mechanism of action might be through the downregulation of the proinflammatory TLR signaling pathway, which decreases the level of proinflammatory cytokines, including IL-6, IL-1β, and TNF-α while increasing that of anti-inflammatory cytokines, such as transforming growth factor-β (TGF-β) and IL-10 [48]. However, the quantitative analysis results on the Shannon index, CRP, TNF-α, IL-10, and IFN-γ revealed negative values, which might be attributed to the inability to stratify different MMAs, given the limited number of outcomes covering the same indicators, each with only two trials examined. They employed different MMAs (varied strains of multistrain probiotics, synbiotics, inulin, and sodium butyrate) targeting distinct mechanisms for gut modulation and T1D amelioration. In this review, one study showed an enriched composition of beneficial gut microbiota, including *Bifidobacterium animalis*, *Lactobacillus salivarius*, and *Akkermansia muciniphila*, and an improved level of TGF-β1 and TNF-α after supplementing with *Lactobacillus salivarius* and *Bifidobacterium animalis* [34]. This was aligned with an intervention study using Jinshuangqi (a triple live probiotic tablet sold in China consisting of *Bifidobacterium longum*, *Lactobacterium bulagricumi*, and *Streptococcus thermophilus*), indicating a decreased level of IFN-γ, Bifidobacterium, and Lactobacillus and a restored Th1/Th2 cell balance in children with T1D [49]. The use of synbiotics (a combination of probiotics and prebiotics) resulted in a significant increase in the levels of SCFAs, ketones, carbon disulfides, and methyl acetates, which was observed to have a greater efficacy on blood glycemic control and inflammation than probiotic usage alone [50,51]. One study in this review indicated a decreased CRP level and an increase in total antioxidant capacity [34].

Sodium butyrate is the most common type of postbiotic, indicating promising glycemic control in streptozotocin (STZ)-induced T1D mice by improving the islet morphology and downregulating the NF-κB-mediated inflammatory signal pathway [50]. In an antibiotic-driven T1D mice model, butyrate ameliorated disease in the female offspring of NOD mice, and in their formal study, butyrate directly shaped pancreatic immune tolerance and dampened T1D progression [51]. Nevertheless, human studies did not support any of these findings [28,29], nor an increase in fecal butyrate. Unlike inulin, a type of prebiotic derived from chicory increased SCFA and interleukin-22 potentially by preventing and/or treating T1D in NOD mice and mitigating symptoms among individuals with T2D through the inhibition of JNK and P38 MAPK pathways [52]. Clinical evidence demonstrated an improvement in gut integrity and higher relative abundance of Streptococcus, *Roseburia inulinivorans*, Terrisporobacter, and Faecalitalea with inulin supplementation in children with T1D [13]. This suggests that the oral intake of postbiotic metabolites from gut microbiota might not act directly and efficiently in promoting the intestinal environment like other supplementations.

Intervention duration and the characteristics of the T1D patients might also play important roles, as revealed in this meta-analysis. MMA intervention for over 3 months demonstrated a significant decrease in HbA1c levels. The between-group heterogeneity for different intervention periods significantly decreased (*I*^2^ = 0% for ≥3 months, *I*^2^ = 8% for >3 months), and the test for subgroup difference reached significance (*I*^2^ = 95.6%), indicating that variations in the intervention period might serve as a probable source of heterogeneity. Different types of MMAs might also be a source of heterogeneity, as indicated by the test in subgroup differences (*I*^2^ = 82.3%). MMA intervention might be more effective in lowering HbA1c in children and adolescents and those with long-term T1D. This could be attributed to the greater adaptability of children’s intestinal flora and the altered glucose metabolism in long-term T1D cases, indicating the increased efficacy of MMA intervention over time [53]. However, the subgroup difference tests revealed that age (*I*^2^ = 31.7%) and disease duration (and *I*^2^ = 0) might not be the source of heterogeneity.

Other potential sources of heterogeneity included dietary factors, physical activity, and the dose and form of MMAs. Only one RCT recorded the dietary factor at baseline and post-intervention, though it reported an unchanged effect after adjusting this confounding factor. The diet also plays a role as studies have shown that HbA1c is lower in patients following a diet with balanced-glycemic-index food [54]. Similarly, a moderate level of PA resulted in better glycemic control in T1D patients [55], but no trials included in this review reported any information on this factor. Lastly, the dose and form of MMAs used may contribute to heterogeneous results, but they are incomparable between different types of MMA, since the unit for probiotics is CFU, while it is gram in postbiotics and prebiotics. While the moderate heterogeneity aligned with the literature [20], the combined effect varied, which might be attributed to the differentiation between T1D and T2D. This review further demonstrated an improvement in DIU and FCP, contradicting the negative results from another review [19]. Furthermore, three studies conducted post-interventional biochemical examinations, revealing no significant difference between the MMA and control groups [15,34,37]. This transient effect aligned with another systematic review, indicating the absence of consistent effects on gut microbiota composition alterations after four to eight weeks of probiotic intervention, suggesting that individuals may require a longer duration of treatment to have therapeutic effects [56].

This is the first systematic review and meta-analysis exclusively focusing on the effect of MMAs and T1D and attempting to investigate inflammation and gut microbiota indicators in addition to glycemic control. However, it is not without limitations. Despite including more than twice the number of studies and sample size compared to previously published research [17,18], the number of ten RCTs was not sufficient for the extrapolation of the results. Meanwhile, even though most of the included studies presented a low or unclear risk of bias, the overall quality and reliability might be compromised due to unclear reporting and missing data. Unclear reporting in studies may lead to difficulties in assessing the true risk of bias, and missing data may result in attrition bias, which may skew the results and reduce the precision of the estimated effects. This review endeavored to quantitatively analyze the gut microbiota composition and inflammatory cytokines post-MMA interventions. However, due to variations in outcome measures, different properties, and targeted outcomes of MMAs, the scope of quantitative analysis was limited, raising the possibility that the insignificant effects on biomarkers other than HbA1c, FCP, and DIU may have been by chance. In addition, statistical heterogeneity was observed in the analyses, confounding factors such as ethnicity differences, dietary factors PA, dose, and form of the intervention were not analyzed in this review.

## 5. Conclusions

In conclusion, this systematic review and meta-analysis suggested that MMA supplementation is associated with improved HbA1c, DIU, and FCP, with moderate quality of evidence in HbA1c and low/very low quality of evidence in FCP and DIU. Multistrain probiotics and synbiotics might exhibit a more significant effect under long-term intervention (<3 months.) Despite the moderate-to-high heterogeneity found in HbA1c and FCP, the evidence supports the potential of MMAs as an adjuvant therapy for glycemic control. The study findings did not substantiate a favorable association between MMA intervention and FBG, CRP, TNF-α, IL-10, LDL, HDL, and the Shannon index, but this might be by chance due to the insufficient number of included studies. Further large-scale clinical trials are necessary to confirm these findings with investigations on inflammation and gut microbiota composition while adjusting confounding factors such as diet, physical activity, and the dose and form of MMA intervention.

## Figures and Tables

**Figure 1 nutrients-16-01675-f001:**
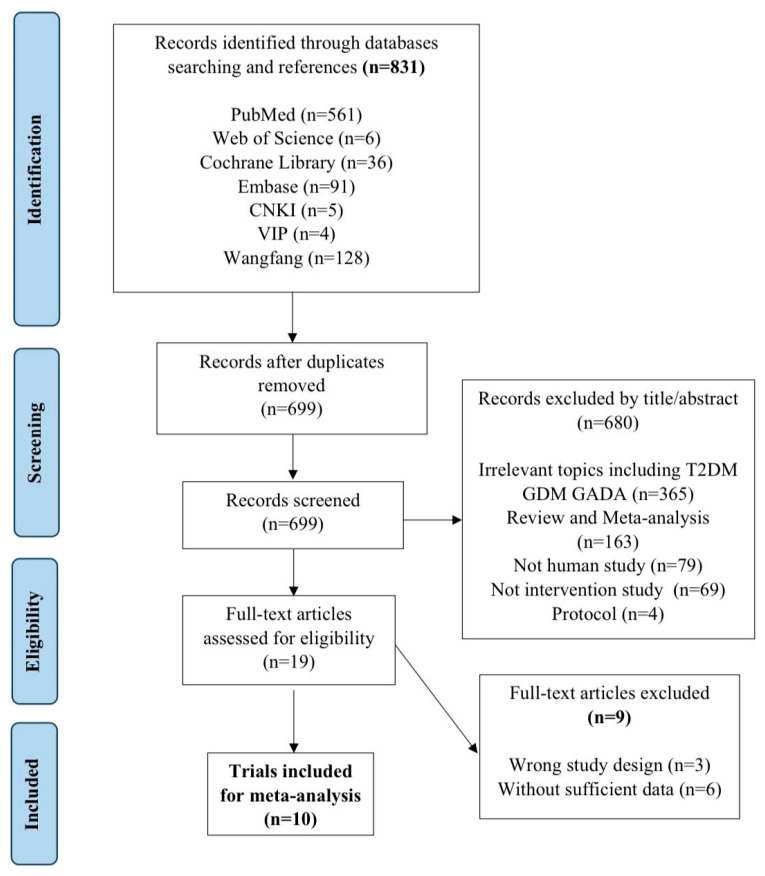
Preferred Reporting Items for Systematic Reviews and Meta-Analyses (PRISMA) flow diagram.

**Figure 2 nutrients-16-01675-f002:**
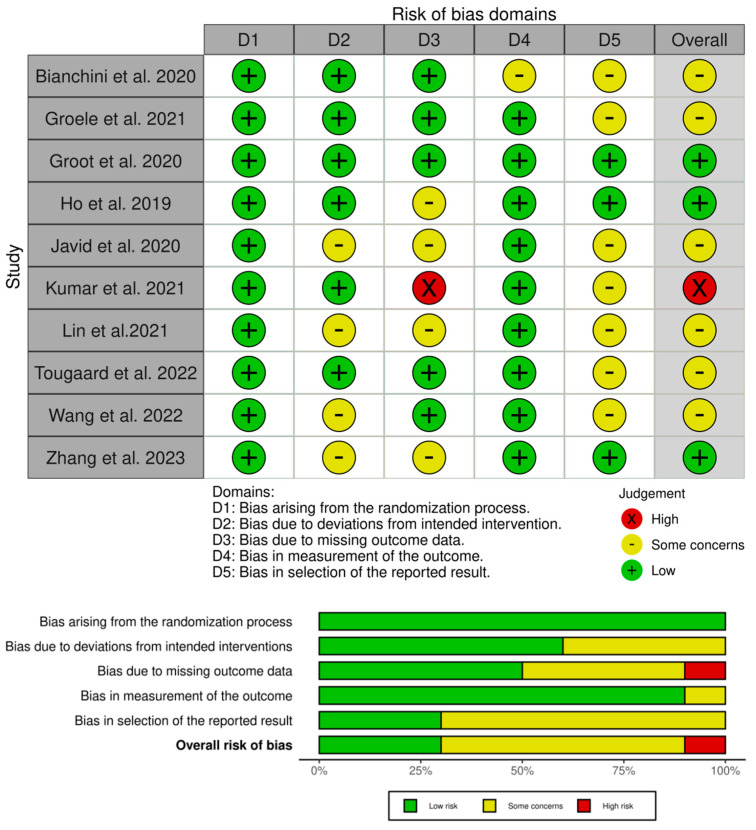
Risk-of-bias assessment for the included 10 studies [15,29,30,31,32,33,34,35,36,37].

**Figure 3 nutrients-16-01675-f003:**
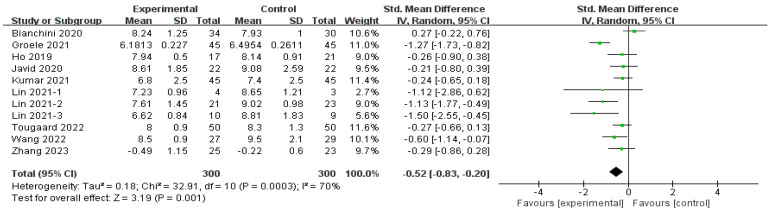
Forest plot of the effect of MMA intervention on HbA1c [15,29,30,31,32,33,34,35,36,37].

**Figure 4 nutrients-16-01675-f004:**
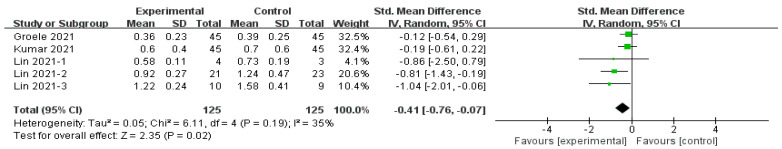
Forest plot of the effect of MMA intervention on daily insulin usage [29,33,37].

**Figure 5 nutrients-16-01675-f005:**
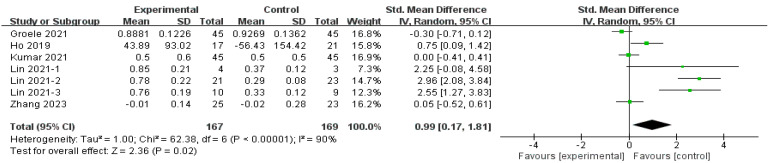
Forest plot of the effect of MMA intervention on fasting C-peptide [15,29,33,35,37].

**Figure 6 nutrients-16-01675-f006:**
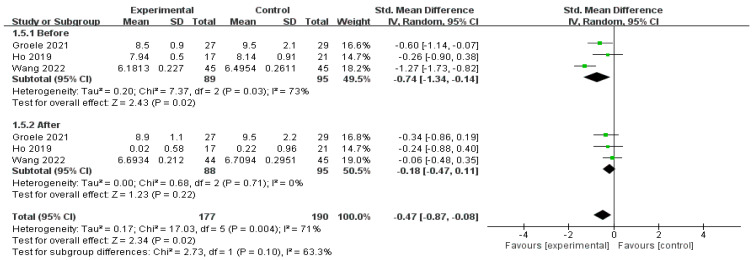
Forest plot of the effect of MMA intervention on HbA1c post-intervention [15,34,37].

**Figure 7 nutrients-16-01675-f007:**
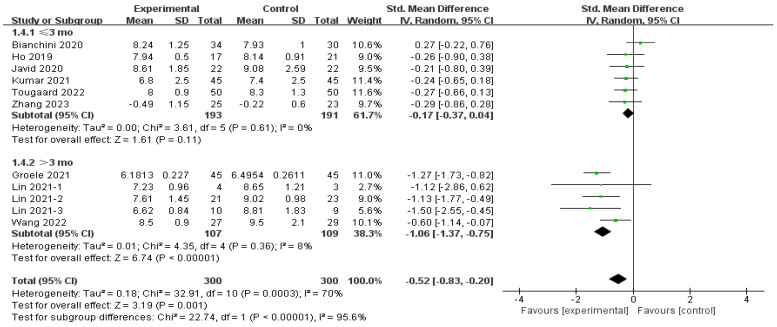
Subgroup analysis of the effect of different intervention durations on HbA1c [15,29,30,32,33,34,35,36,37].

**Figure 8 nutrients-16-01675-f008:**
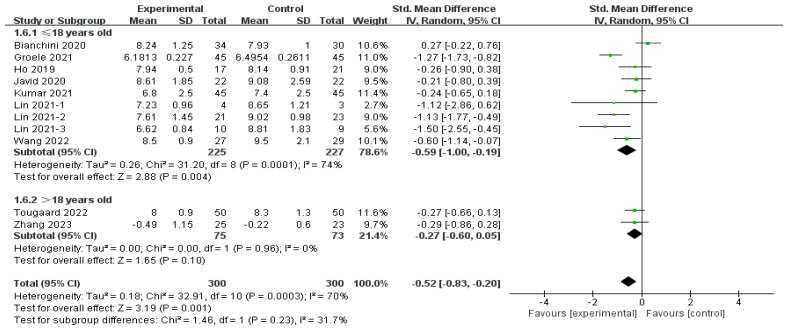
Subgroup analysis of the effect of age on HbA1c [15,29,30,32,33,34,35,36,37].

**Figure 9 nutrients-16-01675-f009:**
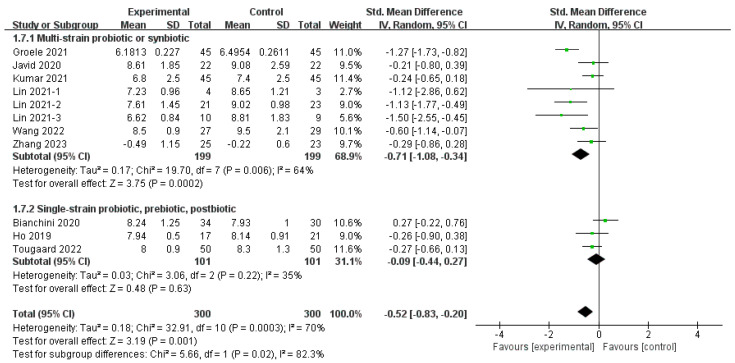
Subgroup analysis of the effect of different MMAs on HbA1c [15,29,30,32,33,34,35,36,37].

**Figure 10 nutrients-16-01675-f010:**
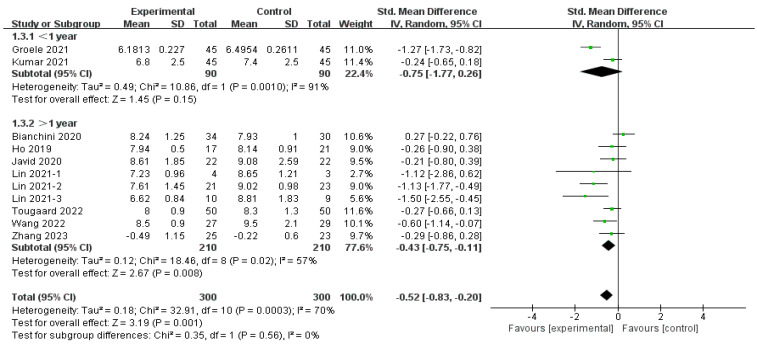
Subgroup analysis of the effect of disease duration on HbA1c [15,29,30,32,33,34,35,36,37].

**Table 2 nutrients-16-01675-t002:** GRADE profile of MMAs for glycemic indices.

Outcomes	Risk of Bias	Indirectness	Inconsistency	Imprecision	Publication Bias	Quality of Evidence
HbA1c	Not a serious limitation	Not a serious limitation	Serious limitation ^a^	Not serious limitation	Not a serious limitation	⊕⊕⊕◯ Moderate
FCP	Not a serious limitation	Not a serious limitation	Serious limitation ^a^	Serious limitation ^c^	Not a serious limitation	⊕⊕◯◯ Low
DIU	Serious limitation ^b^	Not a serious limitation	Not a serious limitation	Serious limitation ^c^	Serious limitation	⊕◯◯◯ Very low

Each circle represents a level of evidence quality: ⊕◯◯◯ indicates very low quality, ⊕⊕◯◯ indicates low quality, ⊕⊕⊕◯ indicates moderate quality. ^a^ There was significant heterogeneity for HbA1c (*I*^2^ = 70%), and FCP (*I*^2^ = 90%). ^b^ Two studies for DIU were evaluated with uncertain risk of bias and one study with a high risk of bias. ^c^ The sample size for FCP and DIU were 336 and 250, respectively, which was less than 400.

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
