# Peer review of "The Effect of Microbiome-Modulating Agents (MMAs) on Type 1 Diabetes: A Systematic Review and Meta-Analysis of Randomized Controlled Trials"

_nutrients, 2024, doi:10.3390/nu16111675_

Round 1
Reviewer 1 Report
Comments and Suggestions for Authors
1. As per the journal's guidelines, the abstract should be around 200 words and without subheadings.
2. The need for an updated review (as noted from previous reviews up to October 2022) is not well explained. In fact there is a newer 2024 review (see: pubmed.ncbi.nlm.nih.gov/38527396) that should have been referenced by the authors and discussed in rationale to the justification for an update review.
3. Suggest to provide the full electronic search strategy used to identify studies, including all search terms and limits for at least one database in the main manuscript. The rest can be provided in the Supplementary Material.
4. The inclusion and exclusion criteria for the review should be better defined. Is there a minimum sample size or study quality threshold in addition to Criteria (5) for primary and secondary outcomes? In terms of interventions, would faecal microbiota transplant or dietary interventions that affect gut microbiome be considered, e.g. yoghurt and dietary fibres? In terms of the exclusion criteria, would other kinds of dietary supplements, nutraceuticals e.g. vitamins and antioxidants also be excluded?
5. How did the authors determine whether "probiotics were taken within three months before the trial"?
6. There are some discrepancies between what is described in the manuscript and what is registered in PROSPERO, e.g. the author list, the inclusion/exclusion criteria for review (e.g. "Individuals are excluded if they have a chronic medical condition that could affect gut microbiota (e.g., Crohn disease, cystic fibrosis, irritable bowel syndrome), or are receiving or received medications or supplements that could affect gut microbiota (e.g., antibiotics, probiotics, prebiotic, laxatives) prior to the study, or had a positive celiac disease screen" which was not mentioned at all in the manuscript) and the data screening/coding steps. Suggest to update what is deposited in PROSPERO to ensure the public record is accurate.
7. Given the very small evidence base for analysis, is there a reason why the authors chose to limit to only RCTs and not clinical trials in general?
8. The methods section mentions using random-effects models but do not explain the rationale for their selection. For example, it should be stated that the random-effects model attempted to generalize findings beyond the included studies by assuming that the selected studies are random samples from a larger population
9. There is considerable heterogeneity in the types of probiotics/product used, patient populations, etc. This was not well-addressed by the authors. There is a strong tendency towards generalizations without adequately addressing inconsistencies or limitations in the current research. More emphasis on the variation in outcomes across different studies and the quality of the individual studies should be more closely scrutinized.
10. Regarding probiotics, the shift in the gut microbiota may be transient and temporary as several treatment trials for probiotics have failed to find significant alterations in gut microbiome; individuals may require longer duration of treatment to have therapeutic effects.
11. "Sensitivity analysis did not show any significant difference in any results" - the sensitivity analysis should have been explained in more detail, particularly with regard to which studies were excluded or included.
12. Figure 11 could be moved to the supplementary material.
13. The authors should more clearly state the level of evidence for at least the main findings (as per GRADE).
14. The conclusion summarizes the findings but overstates the clinical applicability given the methodological limitations and a very small evidence base.
Comments on the Quality of English Language
Extensive edits required.
Author Response
Dear professor,
Many thanks for your comments on our paper, we have revised our paper according to your comments:
Comments 1: As per the journal's guidelines, the abstract should be around 200 words and without subheadings. |
||||||||||||||||||||||||||||
Response 1: Thank you for your comments. The abstract was modified (216 words) and could be found in line 13-28 “Gut microbiome-modulating agents (MMAs) including probiotics, prebiotics, postbiotics and synbiotics are shown to ameliorate type 1 diabetes (T1D) through restoring microbiome from dysbiosis. The objective of this systematic review and meta-analysis was to assess the impact of MMAs on hemoglobin A1c (HbA1c) and biomarkers associated with (T1D). A comprehensive search was conducted in PubMed, Web of Science, Embase, Cochrane Library, National Knowledge Infrastructure, WeiPu, and WanFang Data up to 30 November 2023. Ten randomized controlled trials (n=630) were included, with study quality evaluated using the Cochrane bias risk tool. Random-effects models with standardized mean differences (SMD) were utilized. MMAs supplementation was associated with improvements in HbA1c (SMD=-0.52, 95% CI [-0.83,-0.20]), daily insulin usage (SMD=-0.41, 95% confidence interval (CI) [-0.76,-0.07]), and fasting C-peptide (SMD=0.99, 95% CI [0.17, 1.81]), but had no effects on FBG, CRP, TNF-α, IL-10, LDL, HDL, and shannon index. These findings suggested an association between MMAs and glycemic control in T1D, and subgroup analysis for HbA1c indicated that a longterm intervention (>3 months) might exert a more substantial effect. Further large-scale clinical trials are necessary to confirm these findings with investigation on inflammation, gut microbiota composition, while adjusting confounding factors such as diet, physical activity, and the dose and form of MMAs intervention”. |
||||||||||||||||||||||||||||
Comments 2: The need for an updated review (as noted from previous reviews up to October 2022) is not well explained. In fact there is a newer 2024 review (see: pubmed.ncbi.nlm.nih.gov/38527396) that should have been referenced by the authors and discussed in rationale to the justification for an update review. |
||||||||||||||||||||||||||||
Response 2: Thank you for your comment. The review up to 2022 primarily focused on outcome measures evaluating glycemic control, with no further investigation on inflammatory cytokines. In addition, the study is limited by the number of 5 RCTs. In contrast, this study aims to analyze the outcome measure from the perspective of gut microbiota and inflammation in addition to glycemic control, encompassing double the number of studies and offering deeper insights.
Thank you for sharing the recently published review. After reading this review, we found that while the study covers T1D, it included three trials with T1D patients out of 41 RCTs in total, with a sample size of 83 out of 2991 (2.8%). This review excluded patients under 18, a demographic where T1D is prevalent, and did not differentiate between the two diabetes types which was considered as a limitation mentioned in this review. From the quantity and focus of the included studies, T2D remains the primary concern in this study. Therefore, there is still a gap in quantitative review studies regarding T1D.
The according modification has been made and could be found in line 65-83, “While there is an abundance of reviews exploring the relationship between MMA and glycemic control, the majority of literature predominantly focuses on type 2 diabetes (T2D). Conversely, literature specifically addressing type 1 diabetes (T1D) is notably limited, with only two existing reviews identified. One review, encompassing five randomized controlled trials (RCTs) up to October 8, 2022, examines the impact of probiotics and synbiotics on glycemic control, focusing on outcomes such as fasting blood glucose (FBG), HbA1c, fasting C-peptide (FCP), and daily insulin usage (DIU) [17]. However, it does not delve into outcomes related to T1D pathogenesis, such as inflammatory cytokines and gut microbiota composition. Another recent review aims to explore the effects of probiotic and synbiotic interventions on both T1D and T2D [18]. Despite its inclusion of a large overall sample size, individuals with T1D represented only 2.8% (n=84), and the review excluded patients with diabetes under 18, a demographic where T1D is prevalent. Moreover, this review is unable to differentiate outcomes between T1D and T2D, which is crucial due to their distinct pathophysiologies, treatment modalities, and potential responses to interventions. Consequently, there remains a gap in the literature regarding quantitative review studies on T1D . To address this gap, the current meta-analysis updates the evidence up to November 30, 2023, incorporating ten studies covering children, adolescents, and adults, and employing a more comprehensive set of outcome measures.”
Comments 3: Suggest to provide the full electronic search strategy used to identify studies, including all search terms and limits for at least one database in the main manuscript. The rest can be provided in the Supplementary Material.
Response 3: Thank you for your suggestion. We have made the revision and it can be found in line 96-103, “ A combination of MeSH terms and free text were utilized, encompassing terms such as 'type 1 diabetes,' 'probiotics,' 'synbiotics,' and 'randomized controlled trials.' Boolean operators were employed for sensitivity ('OR') and precision ('AND'), customized to the syntax of each individual database. As an example, the search methodology applied in PubMed was structured as follows: ('Diabetes Mellitus, Type 1'[Majr]) AND ('Probiotics'[Majr] OR 'Prebiotics'[Majr] OR 'inulin'[Majr] OR 'bifidobacterium'[Majr] OR 'lactoccocus'[Majr] OR 'butyrate'[Majr]) with a clinical trial filter. ”
Comments 4: The inclusion and exclusion criteria for the review should be better defined. Is there a minimum sample size or study quality threshold in addition to Criteria (5) for primary and secondary outcomes? In terms of interventions, would faecal microbiota transplant or dietary interventions that affect gut microbiome be considered, e.g. yoghurt and dietary fibres? In terms of the exclusion criteria, would other kinds of dietary supplements, nutraceuticals e.g. vitamins and antioxidants also be excluded?
Response 4: Thank you for your comment. We agree with your point of view that quality threshold should be established for included studies. However, due to the limited number of RCT studies in this field, we opted not to set a minimum sample size or study quality threshold at the beginning (The smallest sample size included in this review had 38 participants, and the overall risk of bias was evaluated as low and unclear).
We have reviewed and revised the overall risk of bias based on the ROB2 handbook (with no changes in individual domains). We found that the quality of several studies was downgraded, but this did not affect the other results. We did a subgroup analysis for HbA1c based on different overall risk of bias, indicating no significant difference. This further indicates that more high-quality RCTs are needed in this field in the future.
Figure 1. Subgroup analysis for HbA1c based on different risk of bias (can be found in the file cover letter)
Considering the dietary factor, the initial idea behind pre-registration in PROSPERO was to encompass all different types of interventions related to improving the gut microbiota. However, upon further investigation, the decision was made to focus on MMA due to its supplemental, medicinal and convenient properties, aligning with the study's aim of providing an economic therapy with high adherence and accessibility for clinical application. FMT surgery, in comparison, is costly and less accessible, with relatively low comparability to oral MMA. Yogurt and dietary fiber are perceived as food items in the public domain. While they may impact gut microbiota, their nature as food products might differs from nutraceutical interventions. Moreover, they involve uncontrollable factors associated with dietary changes and are less comparable to other supplements. Lastly, only one study included mentioned to exclude dietary supplements in the recruiting criteria, and the analysis based on dietary factor indicated no significant alteration. Therefore, FMT, dietary factor and nutraceuticals were not mentioned in the inclusion or exclusion criteria.
Comments 5: How did the authors determine whether "probiotics were taken within three months before the trial"?
Response 5: Thank you for your questions. This information "probiotics were taken within three months before the trial" was a necessary requirement in RCTs that investigated the effect of MMA on T1D, and was mentioned in all RCTs included in this meta-analysis. Thank you for the question, and the team hoped the response answered your question.
Comments 6: There are some discrepancies between what is described in the manuscript and what is registered in PROSPERO, e.g. the author list, the inclusion/exclusion criteria for review (e.g. "Individuals are excluded if they have a chronic medical condition that could affect gut microbiota (e.g., Crohn disease, cystic fibrosis, irritable bowel syndrome), or are receiving or received medications or supplements that could affect gut microbiota (e.g., antibiotics, probiotics, prebiotic, laxatives) prior to the study, or had a positive celiac disease screen" which was not mentioned at all in the manuscript) and the data screening/coding steps. Suggest to update what is deposited in PROSPERO to ensure the public record is accurate.
Response 6: Thank you very much for providing this suggestion, and I apologize for not ensuring the accuracy of this information before submission. The information provided in the manuscript should be considered accurate. The team have already made an updated version under assessment including all the points you mentioned above, and other discrepancies. It could be obtained by the following link: “https://www.crd.york.ac.uk/PROSPERO/display_record.php?RecordID=395896”.
Comments 7: Given the very small evidence base for analysis, is there a reason why the authors chose to limit to only RCTs and not clinical trials in general?
Response 7: Thank you for your question. RCTs provide rigorous control over potential confounding factors, thereby enhancing internal validity and enabling causal inference, which other study designs such as cross-sectional, cohort, and case-control studies often lack. The available evidence in T1D regarding non-RCTs clinical evidence is consisted of approximately 30 studies, primarily focuses on the association between gut microbiota composition and T1D. Nevertheless, the study aim of this review was to investigate whether the intervention of MMA can effectively improve outcomes such as HbA1c in T1D patients. Thus, we have chosen to focus specifically on RCTs in our analysis.
Comments 8: The methods section mentions using random-effects models but do not explain the rationale for their selection. For example, it should be stated that the random-effects model attempted to generalize findings beyond the included studies by assuming that the selected studies are random samples from a larger population
Response 8: Thank you for the comment. We have made the change and could be found in line 162-165, “The utilization of a random-effects model in this analysis served two primary purposes: firstly, it facilitated the extension of findings beyond the included studies by assuming that these studies represented random samples from a broader population; secondly, it was particularly adept at addressing the inherent heterogeneity among the studies.”
Comments 9: There is considerable heterogeneity in the types of probiotics/product used, patient populations, etc. This was not well-addressed by the authors. There is a strong tendency towards generalizations without adequately addressing inconsistencies or limitations in the current research. More emphasis on the variation in outcomes across different studies and the quality of the individual studies should be more closely scrutinized.
Response 9: Thank you for your comment, we have made the change and could be found in line 447-459, “MMAs intervention for over 3 months demonstrated a significant decrease in HbA1c levels. The between-group heterogeneity for different intervention period significantly decreased (I2 = 0% for ≥3 months, I2 = 8% for > 3 months), and the test for subgroup difference was substantial (I2 = 95.6%), indicating variation in intervention period might serve as a probable source of heterogeneity. Different types of MMAs might also be a source of heterogeneity indicated by the test in subgroup difference (I2=82.3%). MMAs intervention might be more effective in lowering HbA1c in children and adolescents and those with longstanding T1D. This could be attributed to the greater adaptability of children's intestinal flora and the altered glucose metabolism in longstanding T1D cases, indicating increased efficacy of MMAs intervention over time [51]. Though, the subgroup difference tests in age(I2=31.7%) and disease duration ( and I2=0) might not be the source of heterogeneity shown in the subgroup difference tests.”
and in line 460-471, “Other potential sources of heterogeneity included dietary factor, physical activity, and dose and form of MMAs. Only one RCT recorded the dietary factor at baseline and post-intervention, though it reported an unchanged effect after adjusting this confounding factor, diet played an role as studies have been shown that HbA1c was lower in patients following a diet with balanced-glycemic-index food[54]. Similarly, moderate level of PA resulted in a better glycemic control in T1D patients [55], but no trials included in this review had reported any information on it. Lastly, dose and the form of the MMAs used might contribute to a heterogeneous result, but it was uncomparable between different types of MMA, since the unit for probiotics was CFU, while it was gram in postbiotic and prebiotic. While the moderate heterogeneity aligned with the literature [18], the combined effect varied, which might attribute to the differentiation between T1D and T2D”.
Comments 10: Regarding probiotics, the shift in the gut microbiota may be transient and temporary as several treatment trials for probiotics have failed to find significant alterations in gut microbiome; individuals may require longer duration of treatment to have therapeutic effects.
Response 10: Thank you for sharing your perspective, we acknowledge and appreciate your points. In this review, only Wang et al. [32] conducted measurement in microbiota composition with an probiotic intervention period for 6 months, suggesting future direction should put into more efforts into trials with longer duration.
Comments 11: "Sensitivity analysis did not show any significant difference in any results" - the sensitivity analysis should have been explained in more detail, particularly with regard to which studies were excluded or included.
Response 11: Thank you for your suggestion. The sensitivity analysis revealed that the omission of any single study did not significantly alter the combined estimate of any outcomes. The modified manuscript can be found in line 230-231: "The sensitivity analysis revealed that the omission of any single study did not significantly alter the result." and line 250-252 “Sensitivity analysis was performed due to high heterogeneity (I2 = 90%, p=0.01, Tau2 = 1.00). An omission of any single study or subgroup did not significantly alter the result, thus subgroup analyses were performed.”
Comments 12: Figure 11 could be moved to the supplementary material.
Response 12: Thank you for your suggestion. Figure 11 was moved to supplementary material and included in Supplementary S3.
Comments 13: The authors should more clearly state the level of evidence for at least the main findings (as per GRADE).
Response 13: Thank you for the suggestion. GRADE was performed following the Cochrane Gradae Handbook (https://gdt.gradepro.org/app/handbook/handbook). We also conducted publication bias for DIU and FCP shown below. We are aware that there should be more than 9 studies to perform funnel plot theoretically, but we came across with high quality meta-analysis which performed funnel plot with 5 studies (10.1016/j.phrs.2022.106399), thus we tried our best to evaluate the result.
The results for the three significant outcome indicators could be found in 1. line 148-152, “The overall certainty of evidence across the studies was graded according to the GRADE guidelines (Grading of Recommendations Assessment, Development, and Evaluation) working group. The quality of evidence was classified into four categories: high, moderate, low, and very low according to the corresponding evaluation criteria”; 2. line 347-370 “3.6 Grading of Evidence An evaluation of the quality of evidence using the GRADE approach was presented in Table 2. The quality of evidence was moderate for HbA1c due to inconsistency (I2=70%), low quality for FCP and very low quality for DIU, owing to limitation on imprecision (n=336 and n=250 for sample size respectively), as well as limitation on inconsistency for FCP (I2=90%) and limitation on risk of bias on DIU and publication bias (two studies had an overall uncertain risk and one was with high risk of bias).
Table 2. GRADE profile of MMAs for glycemic indices. a.There was significant heterogeneity for HbA1c (I2 = 70%), and FCP (I2 = 90%). b.Two studies for DIU were evaluated with uncertain risk of bias and one study with high risk of bias. c.The sample size for FCP and DIU were 336 and 250 respectively which was less than 400..
Figure 2. Funnel plot of FCP Figure 3. Funnel plot of DIU (both could be found in the cover letter)
Comments 14: The conclusion summarizes the findings but overstates the clinical applicability given the methodological limitations and a very small evidence base.
Response 14: Thank you for your valuable opinion, we have made the change and could be found in line 494-506: "In conclusion, this systematic review and meta-analysis suggested that MMAs supplementation is associated with improved HbA1c, DIU, and FCP, with moderate quality of evidence in HbA1c, and low/very low quality of evidence in FCP and DIU. Multi-strain probiotic and synbiotic might exhibit a more significant effect under longterm intervention (<3 months.) Despite moderate to high heterogeneity found in HbA1c and FCP, the evidence supports the potential of MMAs as an adjuvant therapy for glycemic control. The study findings did not substantiate the favorable association between MMAs intervention and FBG, CRP, TNF-α, IL-10, LDL, HDL, and Shannon index, but this might be made by chance due to the insufficient number of included studies. Further large-scale clinical trials are necessary to confirm these findings with investigation on inflammation, gut microbiota composition, while adjusting confounding factors such as diet, physical activity, and the dose and form of MMAs intervention." |

Reviewer 2 Report
Comments and Suggestions for Authors
The review and meta-analysis study of randomized clinical studies is very well conducted, following all the criteria of excellence for carrying out the selection of articles as well as the risk of bias analysis.
Although only 10 studies managed to meet the inclusion criteria, the authors mentioned the limitations of the review in their last paragraph of the discussion section.
Despite not being among the main objectives or outcome of the selected studies. I missed a more detailed analysis of the composition of the microbiota of included participants before and after the MMAs intervention.
Overall the study is very well conducted, opening up the possibility of further RCTs on MMAs in type 1 or type 2 diabetes and also in other metabolic diseases.
Author Response
Dear professor, Thank you for your thoughtful and encouraging feedback on our review and meta-analysis. We appreciate your recognition of the rigorous methodology we employed in selecting articles and conducting the risk of bias analysis.
We also appreciate your suggestion regarding a more detailed analysis of the microbiota composition before and after the MMAs intervention. Indeed, we analyzed the shannon index and obtained an insignificant result. We also discuss about gut microbiota qualitatively in line 413-416 “In this review, one study showed an enriched composition of beneficial gut microbiota including Bifidobacterium animalis, Lactobacillus salivarius and Akkermansia muciniphila , and an improved level of TGF-β1 and TNF-αafter supplementing with Lactobacillus salivarius and Bifidobacterium animalis[32]” and line 437-438 “Clinical evidence demonstrated an improvement in gut integrity, and higher relative abundance of Streptococcus, Roseburia inulinivorans, Terrisporobacter and Faecalitalea with inulin supplementation in children with T1D [13]. ”.
Thank you again for your valuable comments and for acknowledging the potential of our study to pave the way for further RCTs on MMAs in diabetes and other metabolic diseases. We look forward to seeing more research in this promising area. |
Additional clarifications |
We have carefully addressed the recommendations provided by one of the reviewers. While there have been numerous modifications, the deletions have been kept to a minimum. In our efforts to uphold the rigor of the study and ensure the credibility of our findings, we have supplemented additional content in the methods, results, and discussion sections, accompanied by an increase in citations. Additionally, we have revisited and revised the risk of bias (ROB) assessments to ensure a more precise evaluation. Thank you for your assistance and constructive feedback. |

Reviewer 3 Report
Comments and Suggestions for Authors
-Please provide the names of drones in italics throughout the manuscript.
-what exactly did Rayyan do if there were 2 researchers to select the publications?
- .3. Risk of Bias Assessment (it is worth citing which publications were described in which part of the sentence)
-The results of Cochrane's risk of bias assessment in Figures 2 showed that the overall risk of bias was low. Among them, ten studies described the generation of random sequences in detail, six studies (which ones?) reported proper allocation concealment, six studies (which ones?) demonstrated the integrity of data. One study had (which one?) incomplete results due to missing data. In seven studies (which ones?), it was unclear whether there was reporting bias.
-and the same with the remaining subpoints in the results (after 3.3). In section 3.2 it is ok.
-Subsection 3.4 appears twice. Please check if the rest of the numbering is correct.
-I can only partially see Figures 4-6, but maybe it's a system error:( I wrote this problem to the editor.
Author Response
Dear professor, Many thanks for your precious comments on our paper, we have revised our paper according to your comments. Please see the attachment for details of the revisions.
Comments 1: Please provide the names of drones in italics throughout the manuscript. |
Response 1: Thank you for your comment. We have made the modification in line 117-119, “Rayyan is a screening tool used for systematic reviews and meta-analyses, facilitating efficient selection and management of relevant studies [20], and was employed in this review” and line 155-156, “For the synthesis and quantitative analysis of data, Review Manager (Revman) 5.3 software was employed in this study” and
|
Comments 2: what exactly did Rayyan do if there were 2 researchers to select the publications? |
Response 2: Thank you for your comment. Rayyan is a screening tool used for systematic reviews and meta-analyses, facilitating efficient selection and management of relevant studies. We have also accordingly, modified the manuscript in line 117-119 as shown in the first response.
Comments 3: Risk of Bias Assessment (it is worth citing which publications were described in which part of the sentence) Response 3: Thank you for your comments. We modified the manuscript in line 217-222, “The results of Cochrane’s risk of bias assessment in Figures 2 showed that the overall risk of bias was with some concerns. Among them, all studies described the generation of random sequences in detail, six studies reported proper allocation concealment [13,28-31,35], six studies demonstrated the integrity of data [13,28-29,30,32,35]. One study had incomplete results due to missing data [30]. In seven studies, it was unclear whether there was reporting bias [27,28,30-32,34,35,].”.
Comments 4: The results of Cochrane's risk of bias assessment in Figures 2 showed that the overall risk of bias was low. Among them, ten studies described the generation of random sequences in detail, six studies (which ones?) reported proper allocation concealment, six studies (which ones?) demonstrated the integrity of data. One study had (which one?) incomplete results due to missing data. In seven studies (which ones?), it was unclear whether there was reporting bias. Response 4: Thank you for this comment, and we made the change as shown in the above questions.
Comments 5: and the same with the remaining subpoints in the results (after 3.3). In section 3.2 it is ok. Response 5: Thank you for your comments, and we made the change accordingly which could be found in 1. line 228-229 “The efficacy of MMAs on HbA1c was reported by 10 studies (n=600) [13,27,28,30-35,] as depicted in Figure 3”; 2. line 240-241, “The efficacy of MMAs on DIU was reported by 3 studies with 5 subgroups (n=250) [27,31,35]”; 3. line 249-250 “The efficacy of MMAs on FCP was reported by 5 studies with 7 subgroups (n=336) [13,27,31,33,35,]”; 4. line 259-267 “FBG was reported in 3 studies (n=154) [32-34], and the result was not significant (SMD =-0.29, 95% CI [-0.62, 0.03], p=0.08). CRP was reported in 3 studies (n=192) [28,33,34], and the result was not significant (SMD =-0.25, 95% CI [-0.84, 0.33], p=0.40). TNF−ɑ was reported in 3 studies (n=184) [13,32,35], and the result was not significant (SMD =-0.03, 95% CI [-0.58, 0.52], p=0.91). IL−10 was reported in 2 studies (n=128) [13,35], and the result was not significant (SMD = 0.31, 95% CI [-0.04, 0.66], p=0.08). There were 2 studies (n=92) investigated HDL (SMD = 0.27, 95% CI [-0.25, 0.80], p=0.31) and LDL (SMD =-0.23, 95% CI [-0.98, 0.51], p=0.54), and both results were insignificant [33-34]. Shannon index was reported in 2 studies (n=68) [13,29]”; 5. line 271-272 “The efficacy of MMAs on post−interventional result of HbA1c was reported by 3 studies (n=367) [13,32,35]”; 6. line 282-285 “The subgroup analysis involving the intervention duration demonstrated a considerate impact in the subgroup of more than 3 months (SMD =-1.06, 95% CI [-1.37,-0.75], p< 0.01)[27,32,35], and insignificant effect in the other subgroup (SMD =-0.17, 95% CI [-0.37, 0.04], p=0.11)[13,28,30-32,34]”; 7. line 294-296 “Subgroup analysis involving age showed a significant effect (SMD =-0.59, 95% CI [-1.00,-0.19], p< 0.01) in the children and adolescent group [13,27,30,34,31,], whereas this was not observed in adults (SMD =-0.27, 95% CI [-0.60, 0.05], p=0.10) [28,33]”; 8. line 306-310 “The intention was to group each MMA agent respectively, but studies on single-strain probiotic [30], prebiotic [13] and postbiotic [28] having only 1 studies, not allowing for seperate grouping, thus two subgroups were established. Multi-strain probiotics and synbiotic [27,31-34,35] ”; 9. line 320-322 “Subgroup analysis by disease duration showed a significant improvement in longstanding T1D (SMD =-0.43, 95% CI [-0.75,-0.11], p< 0.01) [13,27,28,30,32-34,], and an insignificant effect in onset T1D (SMD =-0.75, 95% CI [-1.77, 0.26], p=0.15) [31,35]”
Comments 6: Subsection 3.4 appears twice. Please check if the rest of the numbering is correct. Response 6: Thank you for this comment. We ensured the numbering is now correct. Apology for this mistake.
Comments 7: I can only partially see Figures 4-6, but maybe it's a system error:( I wrote this problem to the editor. Response 7: Thank you for letting us know. It might indeed be a system error. We contacted the editor, and they have confirmed that "the current version is now visible. |
Additional clarifications |
We have carefully addressed the recommendations provided by one of the reviewers. While there have been numerous modifications, the deletions have been kept to a minimum. In our efforts to uphold the rigor of the study and ensure the credibility of our findings, we have supplemented additional content in the methods, results, and discussion sections, accompanied by an increase in citations. Additionally, we have revisited and revised the risk of bias (ROB) assessments to ensure a more precise and up-to-date evaluation. Thank you for your assistance and constructive feedback.
|

Round 2
Reviewer 1 Report
Comments and Suggestions for Authors
1. The clinical significance and relevance of fasting C-peptide should be at least briefly explained in the introduction.
2. The decision to use a fixed-effects model in cases of low heterogeneity and a random-effects model in cases of high heterogeneity is incorrect. The decision for random-effects or fixed-effects should not be decided based on I2. Instead, it should be decided on the properties of the study and meta-analysis (see: pubmed.ncbi.nlm.nih.gov/28058794).
3. In the assessment for risk of bias, the results section reveals some studies with unclear reporting and missing data, indicating potential bias that is not fully addressed.
4. Regarding probiotics, the shift in the gut microbiota may be transient and temporary as several treatment trials for probiotics have failed to find significant alterations in gut microbiome; individuals may require longer duration of treatment to have therapeutic effects. This should have an accompanying citation (suggestion: pubmed.ncbi.nlm.nih.gov/36986088).
5. The citation for reference 44 is incorrect.
Comments on the Quality of English LanguageModerate edits and wordsmithing necessary.
Author Response
Dear professor:
Many thanks for your comments on our paper, we have revised our paper according to your comments:
Comment 1: The clinical significance and relevance of fasting C-peptide should be at least briefly explained in the introduction.
Response 1: Thank you for your comment. We have revised the introduction and it can be found in line 52-60 “ resulting in increased level of inflammatory substances in the bloodstream[10]. As a result, the inflammation status causes islet auto−immunity leading to decreased fasting C-peptide (FCP) and elevated glycemic level. FCP reflects endogenous insulin production and provides insights into residual beta-cell activity, which is commonly used to assess the effectiveness of interventions aimed at preserving or enhancing insulin secretion [11]. Glycated hemoglobin (HbA1c), a widely used biomarker for assessing long-term glucose control in individuals with diabetes, reflects the average blood glucose levels over the past 2-3 months, providing information about the effectiveness of diabetes management strategies [12]”.
Comment 2: The decision to use a fixed-effects model in cases of low heterogeneity and a random-effects model in cases of high heterogeneity is incorrect. The decision for random-effects or fixed-effects should not be decided based on I2. Instead, it should be decided on the properties of the study and meta-analysis (see: pubmed.ncbi.nlm.nih.gov/28058794).
Response 2: Thank you for your comment. In addition to the literature you suggested, I referred to the Cochrane Handbook and another paper to gain a deeper understanding of the differences between the fixed-effects and random-effects models. I have made the necessary corrections to the manuscript accordingly.
Please see the changes on line 169-175 “The random-effects model posited that the treatment effect estimates observed in studies may vary due to genuine disparities in treatment effects across each study, along with sampling variability. This diversity in treatment effects could be attributed to discrepancies in study populations (e.g., patient age), interventions administered (e.g., drug dosage), duration of follow-up, and other variables. Thus random-effects model was utilized by facilitating the extension of findings beyond the included studies by assuming that these studies represented random samples from a broader population.” I sincerely appreciate the valuable guidance. Thank you for your assistance.
- https://training.cochrane.org/handbook/current/chapter-08#_Ref396934606
- https://www.bmj.com/content/342/bmj.d549
Comment 3: In the assessment for risk of bias, the results section reveals some studies with unclear reporting and missing data, indicating potential bias that is not fully addressed.
Response 3: Thank you for your comment. We acknowledge that unclear reporting and missing data in some studies can introduce potential bias. Since there were more than half studies with unclear reporting and missing data, we were unable to perform sensitivity analysis, thus we did two subgroup analyses based on the two domain and yielded insignificant results. Therefore, we have addressed these issues in the limitations sections.
For detailed information, please refer to line 494-499 “ even though most of the included studies presented a low or unclear risk of bias, the overall quality and reliability might be compromised due to unclear reporting and missing data. Unclear reporting in studies might lead to difficulties in assessing the true risk of bias, and missing data might result in attrition bias which might skew the results and reduce the precision of the estimated effects.
Comment 4: Regarding probiotics, the shift in the gut microbiota may be transient and temporary as several treatment trials for probiotics have failed to find significant alterations in gut microbiome; individuals may require longer duration of treatment to have therapeutic effects. This should have an accompanying citation (suggestion: pubmed.ncbi.nlm.nih.gov/36986088).
Response 4: Thank you for your comment and suggestion. We have made the change, please see line 486-489 “This transient effect aligned with another systematic review indicating a dearth of consistent effect on gut microbiota composition alterations after four to eight weeks of probiotic intervention, suggesting individuals may require longer duration of treatment to have therapeutic effects [56].”
Comment 5: The citation for reference 44 is incorrect.
Response 5: Thank you for your comment. We checked with citation 44 (which is now citation 46) and made corresponding modification, please refer to line 647-648 “46.Chapman CM, Gibson GR, Rowland I. Health benefits of probiotics: are mixtures more effective than single strains?. Eur J Nutr. 2011;50(1):1-17. doi:10.1007/s00394-010-0166-z”
We would like to extend our sincere gratitude to you for your valuable comments and suggestions, which have significantly contributed to the improvement of our manuscript. We appreciate your time and effort in reviewing our work and providing such insightful feedback. Thank you for your consideration and support.